# The limitations of automatically generated curricula for continual learning

Anna Kravchenko[1]*, Rhodri Cusack[2]

**1** Faculty of Science, Radboud University, Nijmegen, The Netherlands, **2** Trinity College Institute of Neuroscience, Trinity College Dublin, Dublin, Ireland

* anna.kravchenko@ru.nl

**Data Availability Statement:** All data and code available from a public repository: https://github.com/ankravchenko/curriculum_learning.

**Funding:** This study was supported by the European Research Council Advanced Grant (https://erc.europa.eu/): 787981 (RC) and Science

## Abstract

In many applications, artificial neural networks are best trained for a task by following a curriculum, in which simpler concepts are learned before more complex ones. This curriculum can be hand-crafted by the engineer or optimised like other hyperparameters, by evaluating many curricula. However, this is computationally intensive and the hyperparameters are unlikely to generalise to new datasets. An attractive alternative, demonstrated in influential prior works, is that the network could choose its own curriculum by monitoring its learning. This would be particularly beneficial for continual learning, in which the network must learn from an environment that is changing over time, relevant both to practical applications and in the modelling of human development. In this paper we test the generality of this approach using a proof-of-principle model, training a network on two sequential tasks under static and continual conditions, and investigating both the benefits of a curriculum and the handicap induced by continuous learning. Additionally, we test a variety of prior task-switching metrics, and find that in some cases even in this simple scenario the a network is often unable to choose the optimal curriculum, as the benefits are sometimes only apparent with hindsight, at the end of training. We discuss the implications of the results for network engineering and models of human development.

## 1 Introduction

Human education is a refined enterprise. Those around a child adapt their behaviour in many ways to help the child's learning. In preschool and school, teachers follow carefully structured curricula, designed to introduce foundational concepts before more advanced ones. Pedagogy is a science in itself and we dedicate no small amount of effort to finding out how to teach children in an optimal way. Some studies suggest that even slow biological maturation of infants' cognitive abilities leads to better learning [1–4].

Curriculum learning experiments, starting from Elman's classic work [1], have found that artificial neural networks (ANNs) can also benefit greatly from elaborate approaches to training. Starting with more representative examples, and gradually increasing in diversity, improved convergence speed and helped the network to focus on relevant features while it was forming representations [5]. Using auxiliary tasks has proven to be effective in improving

Foundation Ireland (https://www.sfi.ie/): 17/RC-PhD/3482 (RC). The funders had no role in study design, data collection and analysis, decision to publish, or preparation of the manuscript.

**Competing interests:** The authors have declared that no competing interests exist.

generalization across tasks and robustness against various types of noise [6–8]. Recent experiments with ANN training datasets have also demonstrated that some ANN limitations that are considered an inherent property of ANN architectures, such as, for example, a bias towards local processing, can be removed by adjusting the dataset [9].

When training a network to perform multiple tasks a correctly chosen curriculum has been shown to push the network towards either more specialized or more flexible representations without having to modify the architecture [10], affecting the emergence of alternative neural mechanisms and allowing the researchers to choose between higher performance for learned tasks at the costs of flexibility or a better performance over a wide range of tasks at the expense of a lower performance for the learned tasks.

Decomposing a complex task into a sequence of source tasks with gradually increasing complexity has been shown to greatly improve performance for certain types of reinforcement learning [11].

Curriculum learning has also been proven especially useful when dealing with continual learning. Modern machine learning excels at training powerful models from fixed datasets and stationary environments but these models often fail to emulate the robustness and efficiency of human learning in a non-stationary world [12, 13]. One of the most known cases is catastrophic forgetting—the rapid performance degradation on earlier learned tasks that occurs when dealing with highly non-stationary data—but ANN models more broadly under-perform when presented with changing or incremental data regimes [14]. A solution proposed for this problem is using a curriculum, including using auxiliary tasks [15], pre-training [16, 17] and active learning [18, 19].

The idea of active learning (also known as curiosity-driven learning and automated curriculum learning) initially stemmed from addressing a problem that using curricula introduces: designing an effective curriculum can be hard as there are many free parameters. It therefore requires time and effort to create and it may not generalise to other training sets. Instead of manually creating curricula, an attractive alternative is that the network, by monitoring its learning, could automatically choose its own curriculum—automated curriculum learning. This approach is appealing as it makes design simpler and could allow networks to adapt to new datasets and continuous streams of data.

This applies to sequences of tasks too. Graves et al [20] showed that automated curriculum learning can indeed significantly accelerate progress and increase accuracy by proceeding from easier to harder tasks. Extensive experiments on the Nevis'22 Benchmark [21] have also demonstrated that randomizing the order of the tasks in a continual stream of multiple tasks had dramatic effects on performance. However, the method proposed by Graves requires a suitable progress metric used to indicate when to switch tasks. The best metric varied for different sets of tasks, which raises the question under which conditions actively choosing a curriculum for learning is possible, and whether it is achievable to identify the best progress signal in advance, without retraining the network for every potential metric.

This is a particular concern in developmental robotics, a scientific field which, among other things, aims at studying the developmental mechanisms and patterns by building robots able to learn from their surroundings and choose their own training input. Curiosity-driven algorithms have been shown to encounter pathological cases, such as becoming fixated on learning from completely unpredictable stochastic data [22] and in developmental robotics they can fail when the need to switch between a variety of tasks arises [23]. Choosing a stream of input relying on curiosity alone doesn't seem to be a viable strategy.

The literature on human learning is also relevant. In education there has long existed a conflict between curiosity-driven learning and fixed curriculum learning [24]. Curiosity has been demonstrated to enhance learning, both by improving memory [25, 26] and by helping

students to learn important background information by deviating from a fixed curriculum [27]. Children and infants are drawn to things that have just the right "Goldilocks" level of novelty [28, 29], which has been shown to be a strategy of creating an optimized learning curriculum that is also applicable in machine learning [30]. External incentives have been shown to suppress curiosity-driven intrinsic motivation in some cases and harm learning [31]. However, at the same time, in human learning, formal schooling, which to a large extent uses a fixed curriculum is highly effective, increasing the intelligence of learners by 1–5 IQ points per year [32]. When dealing with difficult tasks humans often need at least some degree of guidance, especially in infancy, when their knowledge of the world is limited. Machines are not the only ones who struggle with "pure" active learning.

The need for guidance is further supported by the Vygotskian concept of scaffolding [33]. By providing support in tasks that are just above the student's skill level and breaking tasks into small manageable chunks a competent teacher can allow them them to learn content they wouldn't have been able to process on their own [34]. Our hypothesis is that ANNs may require very much the same treatment.

The cumulative evidence thus indicates that we need a better understanding under which exact conditions automatic curriculum building/curiosity driven learning is viable. The goal of this paper is to investigate such conditions, as well as the optimal metrics to guide learning.

In continual learning in ANNs, an assortment of goals needs to be achieved, including avoiding forgetting, learning sequences of tasks, and forward knowledge transfer. In this paper we address the problem of knowledge transfer and the problem of switching between tasks (i.e. assessing when a supporting task has been sufficiently 'learned').

To investigate this, we created a network and a task for which curriculum learning was important, evaluated a variety of training regimes and measured whether an optimal regime could be discovered by the network itself.

## 2 Methods

### 2.1 Data and tasks

Our goal was to design a task set that would benefit from training with a curriculum, but that was sufficiently simple to allow exploration of final performance across many training scenarios. The curriculum of our network consisted of two tasks. Task 1 was a simple supporting task of recognizing single-digit numbers. There was then one of two more complex main tasks. Task 2a required the network to discriminate whether the sum of two digits in the picture was odd or even. Task 2b (discussed further in section 2.3.2) required the network to calculate the sum of two digits in the picture.

Task 2a could be conceptualised as two tasks that we would expect the network to be able to learn in isolation: an odd/even classification for each of the digits (which should be of a similar difficulty to digit recognition and would require the network to create single digit representations); and XOR to resolve the digit sum—*even* if both single digits are either odd or even, and *odd* if one is odd and the other even. While the network should be able to learn each of these tasks in isolation, when they are presented together, we predicted the network would fail to learn. Each digit is equally distributed among the odd-even digit-sum classes and so the error signal would be orthogonal to the presence of any given digit. We therefore predicted the network would be unable to learn the single digits (or the initial odd-even classification). Put another way, during training the weight changes each batch will cancel each other out (assuming the XOR transformation is not yet present at the higher level), which would make it difficult to learn the image-to-single digit mapping required for good task performance. We therefore predicted that pre-training on Task 1 would be a valuable stage towards optimizing

performance on Task 2a, as learning the digit representations beforehand eliminates this problem. In this simple model a similar result could potentially be achieved by optimising the amount of training epochs or modifying the architecture of the network. However, in real life applications and with more complex tasks with similar properties this could require significant computational resources to adjust the network in an effective way. As the The Nevis'22 Benchmark experiments [21] show, the assessment of the compute spent while learning should include the compute spent while doing hyper-parameter search to evaluate the models effectively. This makes the optimization provided by the curriculum a welcome addition.

More generally, we hypothesized that such hierarchical learning scenarios will be encountered when dealing with tasks that require higher-level abstract information (such as learning language or mathematical equations) or where sub-goals seemingly require moving further away from the final goal (the Tower of Hanoi game). However, some abstract tasks may not have this property. To investigate this, we explored a variant, Task 2b, where the network had to learn the sum of digits in the picture instead of whether that sum was odd or even. Our prediction was that, although this is also a multi-digit task building upon single-digit representations with an even greater number of potential answers, it would be more easily learnable as the error signal for the multi-digit task was not orthogonal to the single-digit classes. We hypothesized that the Task 2b error signal will thus be able guide effective learning of the single digits.

We evaluated a pre-training curriculum, where Task 1 always preceded Task 2, with varying degrees of training on each.

The training datasets were based on the MNIST dataset [35]: a set of 60000 28x28 greyscale images of handwritten digits, centered and normalized. In both tasks, the network was trained in a supervised way (Fig 1). For the "simple" Task 1, digits were scattered on a white background 84x84 squares to ensure learning spatially invariant representations Note the images were padded and not resized, and so scale invariant training was not needed). The network was required to recognize single digits. For Task 2, we used MNIST dataset to create 55 classes of double digit stimuli, comprising every combination of two unordered digits. The digits were again scattered on the 84x84 white background and were constrained so that they never overlapped. Both sets consisted of 30000 examples each. The test set consisted of 10000 double-digit images labeled for performing an odd-even task.

## 2.2 Model

To best explore the benefits of curricula, we used a simple ANN, with two convolutional layers (kernel size = 3, ReLU activation, average pooling. the first layer was comprised of 64 filters and the second one of 32 filters) followed by a densely connected layer of 10 nodes, and two dense layers of 2 nodes (Fig 1, centre column). During Task 1, the loss function was calculated from the first fully connected layer with 10 nodes. During Task 2 the loss function was calculated at the final layer. This simple network did not perform at ceiling on the target task, allowing us to distinguish the relative strengths of different curricula. There are architectures that could allow better performance in Task 2a/b, irrespective of the training regime. However, to provide sensitivity to the training manipulations, we intentionally chose network architecture and hyperparameters that did not perform at ceiling on the task (see S1 Fig, in the Supporting information section).

Adding a second output layer that follows output layer 1 for Task 2 was also dictated by the performance not being at ceiling. As we used a network with only 2 convolutional layers, we were cautious of the possibility that proper digit representations might only be formed in the

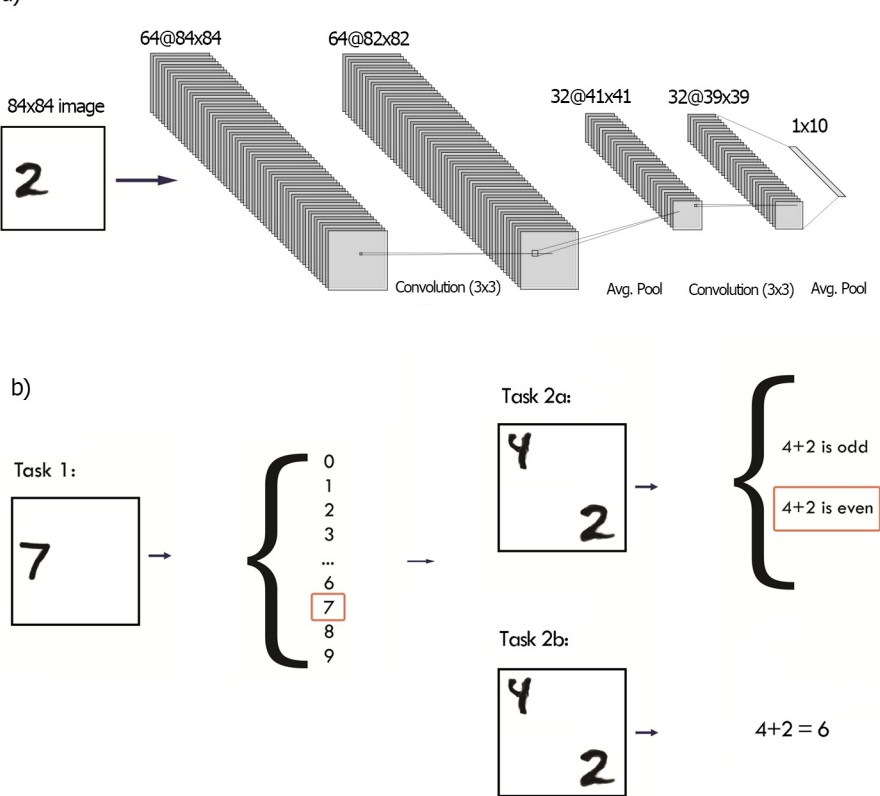

**Fig 1. Architecture and tasks.** Task 1: Single digit recognition task. Task 2a: Sum of digit odd/even recognition task. Task 2b: Sum calculation task. In all tasks digits were scattered on a white background to ensure learning spatially invariant representations.

final dense layer and thus removing them would further hinder network's ability to learn Task 2.

To investigate the generalisation of switching metric performance across networks, we also tested a network variant that was similar, except that it had three rather than two convolutional layers.

The models were implemented using python 3.5 and Tensorflow 2.0, and ran on an Amazon Web Services g3s.xlarge machine (4 virtual CPU, 30.5 GiB RAM, Tesla M60 GPU). The code can be found at https://github.com/ankravchenko/curriculum_learning.

If a metric for curriculum learning is going to be useful, it has to be informative within a single instance of training a network. We therefore evaluated instance-to-instance variation, by training each model 8 times with random starting weights, to obtain a mean and standard deviation of performance accuracy and switching metrics. For calculating these values and further statistical analysis the Python SciPy library was used (version 1.7).

## 2.3 Training procedure

**2.3.1 Comparing static and continual learning.**  Both the Task 1 and Task 2 stimuli were constructed from the 30000 MNIST digits, with 3000 examples for each of the digits 0–9 and for 250 writers [35].

For static pre-training, a subset of $n * 1250$ digits was sampled without replacement from the 30000 digits, where $n = [0 .. 24]$ determined the quantity of pre-training. In each of three epochs, the entire subset was presented in random order in batches of 32.

For continual pre-training, $n$ non-overlapping subsets of 1250 exemplars were sampled without replacement from the pool of 30000 MNIST digits. These subsets will have on average sampled the digits and the writers in a balanced way, but differed in their specific exemplars. Three epochs of training were conducted on the first subset, then three epochs on the second, and so on, until $n$ subsets had been used. This "exemplar incremental" design reflects a weaker form of non-stationarity than is often present in the environment, but it still affected network learning substantially.

Task 2 training then followed. For comparability, the same procedure was used for static and continual learning. 24 non-overlapping subsets of 1250 exemplars were sampled without replacement from a pool of 30000 images. Three epochs of training were conducted on the first subset, then three epochs on the second, and so on, until $n$ subsets had been used.

At no point in the training process were the weights frozen. Adam optimizer was used for gradient descent, with a constant learning rate of $10^{-3}$, no regularizer was used in any of the layers. The choice of hyperparameters, as well as the small number of epochs, were dictated by the need to limit the learning capabilities of the network, making the task only barely learnable to investigate the effects of curriculum.

**2.3.2 Testing the orthogonality hypothesis.** To test our prediction about the effect of the orthogonality of the error signal, we also included a variation of the second task, Task 2b. The image stimuli of double digits remained the same, the labels, however, were changed to a sum of digits present on the picture.

To ensure that the labels were not orthogonal to the component digits, we provided the network with a "magnitude style" label encoding:

$$l_i = \begin{cases} 1, & \text{if } i \le X \\ 0, & \text{if } i > X \end{cases}$$

where $l_i = [1 .. 20]$ is the output of neuron $i$, and $X$ is the sum of the two digits.

As the labels were no longer "hot one", the softmax layer was removed and the loss function was altered from classification accuracy to the mean absolute error.

Since we were only interested in learnability, for Task 2 the training procedure was also simplified. We used static training and investigated three different amounts of pre training: for 30000, 15000 and 0 images of Task 1, with training on the full set of Task 2 after that.

## 2.4 Metrics to predict an optimised switching moment

We then investigated under what conditions the network itself could determine the optimal moment for switching between tasks using only the data available during a single training run. Here we define the optimal switching moment as the amount of pre-training on Task 1 needed for the network's mean final performance on Task 2 (averaged across 8 runs) to reach the top decile of the accuracy observed over all regimes of training.

In the existing literature a variety of metrics have been used to help the learner to determine when to switch between task stages. Oudeyer [23] called them adaptive intrinsic measures and categorised them into competence-based, predictive and information-theoretic types. Graves [20] added complexity gain measures (effective information encoded in the network weights) to the list. Based on this, we considered four kinds of promising metrics:

- **Performance**. A popular approach in curriculum learning is to define a performance threshold on the pre-training task for advancement to the next task [36]. We therefore investigated if there was a threshold, consistent across scenarios, to define the optimal switching level.

- **Competence**. A typical competence measure would be to evaluate initial Task 2 performance (performance after the first training set slice of 1250 examples on Task 2 after pre-training on Task 1) after a given amount of pre-training.

- **Prediction gain**. In the literature on human learning, intrinsic motivation is viewed as a system that generates rewards when predictions improve over time. Such a system will try to maximize learning/prediction progress, i.e., to decrease of prediction errors. To capture this, the gradients of performance on Task 1 and Task 2 were captured by the change in performance across a step of 1250 trials as measures of learning progress to be used for determining the switch time.

- **Information gain**. Information theory has also been used in automated curriculum learning. Building on the work of [37], we hypothesized that the peak in Fisher Information (FI) of weights, which corresponds to a phase transition in learning, might be an effective signal that could guide the switch in learning. We evaluated the Fisher Information for the weights calculated with the loss functions from both Task 1 and Task 2. Both the FI peak and the post-peak drop could be viable points of switching, therefore to determine the optimal point we calculated Pearson correlation coefficient between FI value at the moment of switching and the final performance on Task 2 at each n*1250 examples step of pre-training on Task 1 using information from 8 instances of our network. For each 1250-example step of pre-training on Task 1 we also compared FI for the possible next steps: another 1250-example slice of training on Task 1 and first 1250-example slice of Task 2, as greater information gain from a different task could also indicate a viable switching point.

## 2.5 Examining task representations

To investigate how representations developed in the network during pre-training phase, we used t-SNE analysis to visualise activations in the first dense layer of the network at various stages of pre-training on Task 1 in response to selected diagnostic classes, using 100 examples of each class. The single digits 1, 3 and 8, and their double-digit combinations (11, 13, etc) were used as diagnostic classes, selected on the basis of 3's similarity to 8 and dissimilarity of 1 to both of the former.

Implementation of t-SNE algorithm from sklearn python library was applied using 2 dimension components, 300 iterations, with perplexity to 40 and verbosity level set to 1. No other dimensionality reduction methods were used before feeding the data to t-SNE.

We also displayed activations for ten random examples of each image class to explore individual variation in embeddings.

## 3 Results

### 3.1 Testing a range of curricula

Our first goal was to investigate if the network benefited from training on Task 1 first. Both for continual and static learning, final performance on Task 2 depended on the amount of pre-training on Task 1 (Fig 2), with Task 2 unlearnable without some initial training on Task 1. This shows that this task structure was effective in requiring a curriculum. For continual

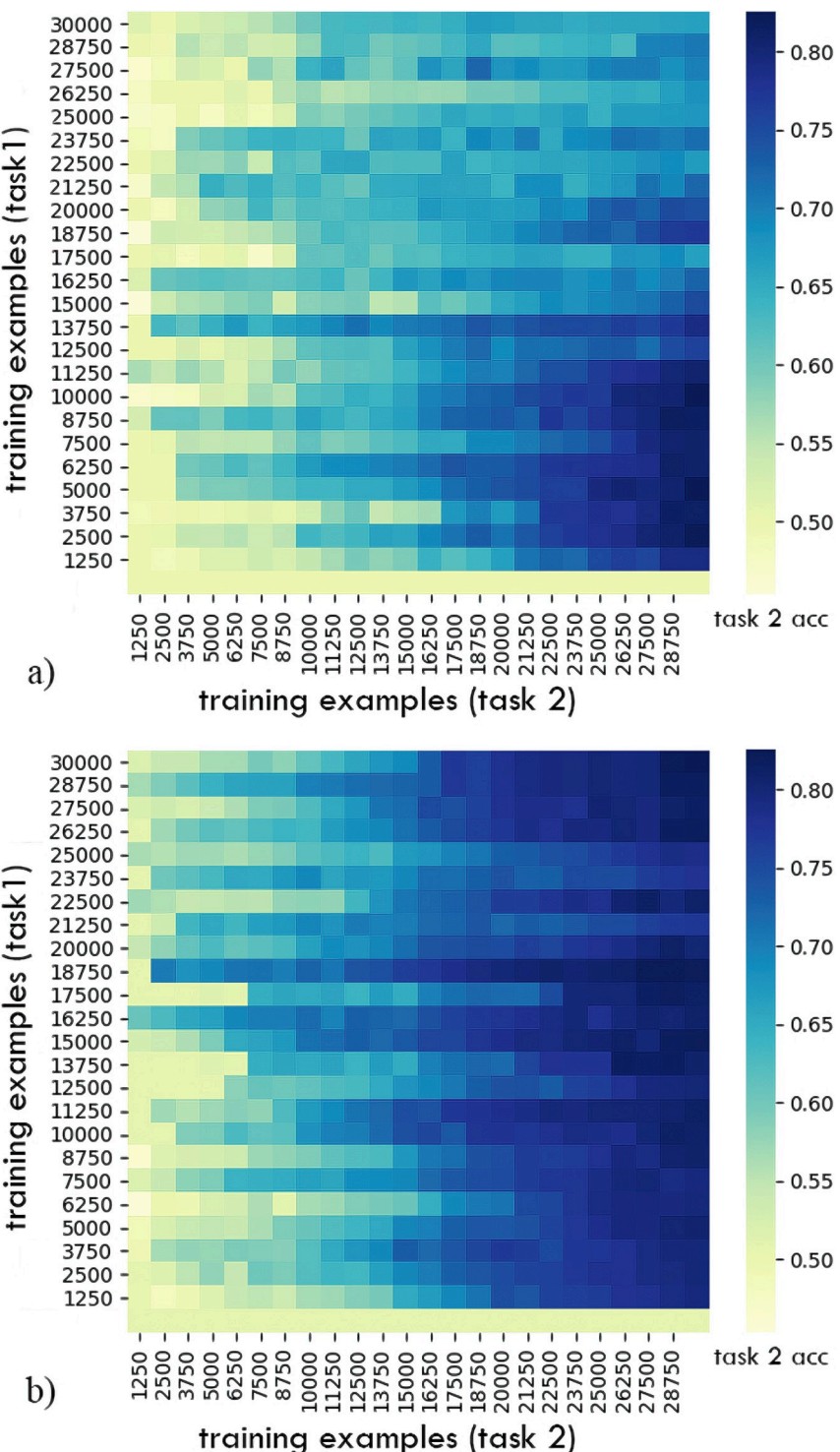

**Fig 2.** Task 2 performance for continual (a) and static learning (b) of Task 1. The average of 8 runs is shown. With no Task 1 pre-training (bottom row of either panel), Task 2 performance was at chance irrespective of the quantity of Task 2 training. This confirms the task structure was effective in requiring a curriculum. With continual learning, too much Task 1 pre-training was also detrimental to final Task 2 performance. This could to overfitting and accumulation of error when dealing with a sequence of tasks.

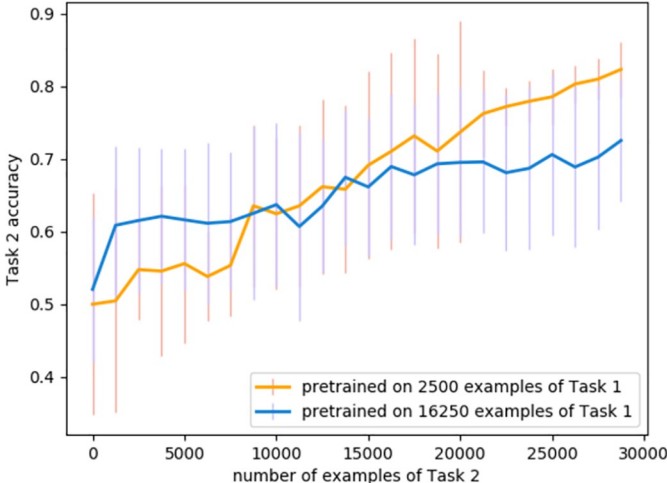

**Fig 3. Comparison of learning trajectories for Task 2 after different amount of pre-training.** During continual learning, in single runs the effect of too much pre-training on Task 1 on Task 2 performance did not emerge until later in the Task 2 training.

learning, too much pre-training could also be detrimental (Figs 2a and 3). Thus, the network benefited from having a curriculum, and the amount of training on Task 1 was important.

It is worth noting that simply splitting the curriculum in two tasks without learning them in the right order didn't solve the problem. Learning the XOR task first was not beneficial either: the performance was still no better than random (Fig 4).

## 3.2 Metrics to predict the optimised switching moment

To optimise the curriculum, we evaluated four types of metric that could be used by the network for task-switching. We discuss each in turn.

- **Performance**. When tested on the original 2 convolutional-layer network with a standard learning rate performance on Task 1 seemed a possible indicator of the optimal switching moment during continual learning (Fig 5a, orange). To test if the critical threshold of performance generalised across scenarios, we repeated training and analysis for a variant on this ANN, with three instead of two convolutional layers. The Task 1 performance level that signalled the optimal switching moment was not consistent across these two networks, ruling out this as an effective metric (Fig 5). The same was observed during stationary learning: as can be seen at Fig 6, the red line marking to the optimal switching moment doesn't correspond to any visual cues or irregularities in any of the metrics.

- **Competence**. Initial competence on Task 2 showed no consistent change as a function of the quantity of Task 1 training (Fig 5b, blue), and so this will not be useful as a metric. Furthermore, even after a little training, initial competence on Task 2 was not informative, because of pre-training sometimes yielding delayed benefits that were not apparent until considerable Task 2 training (Fig 3).

- **Prediction gain**. Prediction gain metrics did not provide a possible signal for switching in our tasks either. While at some points of training Task 2 "learning progress" exceeded Task 1 learning progress, it was indistinguishable from noise (Fig 5 1b). Even if we were to attempt

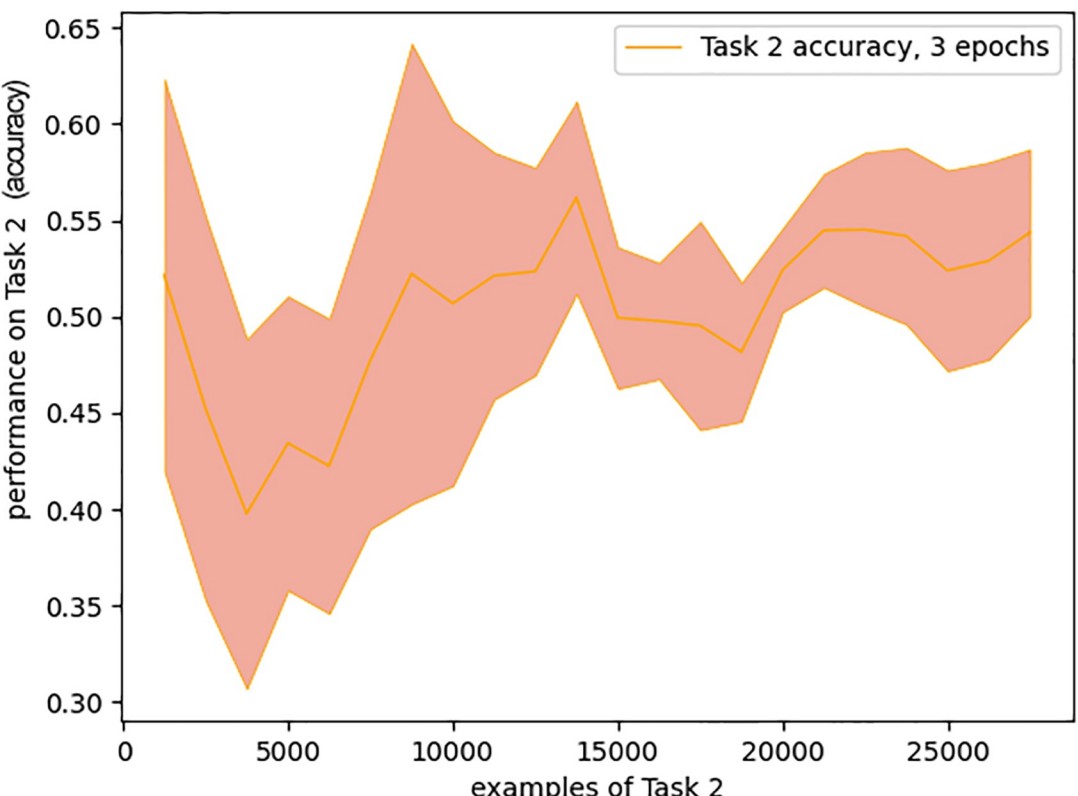

**Fig 4. Learning Task 2 first followed by Task 1.** Learning the XOR task first did not lead to an improvement in performance.

to use learning progress as a possible measure, as it showed substantial run-to-run variation and in many cases it would switch too early.

- **Information gain**. For Task 2 there was also no statistically significant difference between Fisher information (FI) of weights at optimal and suboptimal points of switching (Table 1). This can be explained by the requirement to learn higher-level features to perform Task 2, and is in line with [37] finding that high-level features are not affected by critical periods. The FI of weights in Task 1 was partially effective at predicting the beginning of switching period in our original network. To explore whether this decrease in weight-change generalised to other cases of learning, and whether this effect may arise due to switching from feature memorizing to feature reorganization described by Achille [37], we calculated FI for Task 1 for a deeper network and for the original network trained on smaller slices. As can be seen on Figs 5 2c and 7, FI in these tasks showed no correlation with final performance only peaking after the optimal switching window in these cases (Fig 8). FI for the next step of training on Task 1 also remained significantly greater than FI after switching to Task 2 (Fig 9). This shows that FI is not a reliable indicator to guide curriculum learning.

In sum, we found that neither main nor supporting task performance or their gradients, nor Fisher information of weights provided a signal that could guide active learning when training a network in a continual way. These results provide no evidence that the optimal learning strategy indeed can be derived from information available to the network.

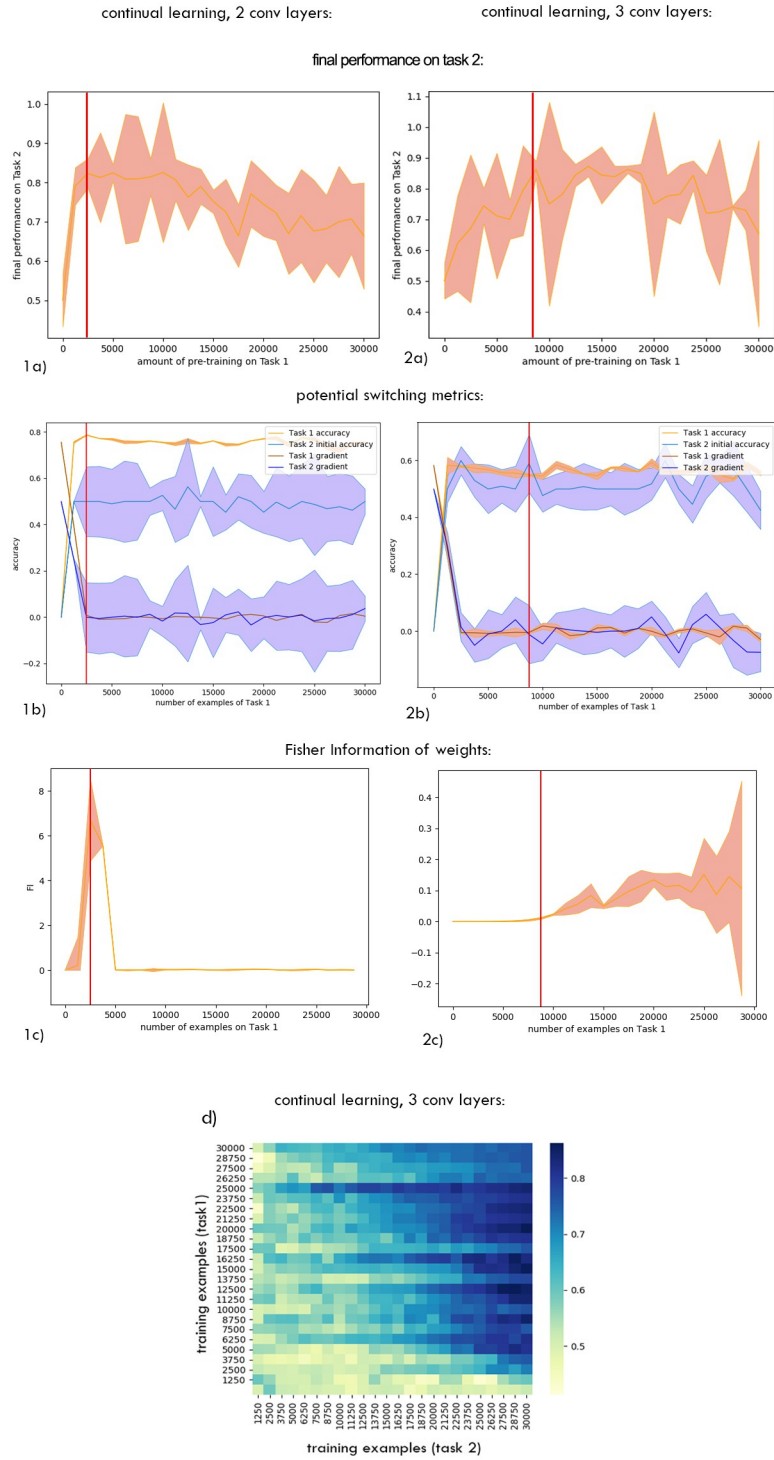

**Fig 5. Comparison of potential switching metrics during continual learning.** Network versions with 2 and 3 convolutional layers were compared. On all figures red vertical lines indicate the optimal switching window. a) Mean of 8 runs and standard deviation bands for final performance on Task 2 with respect to the amount on pre-training on Task 1. b) Neither auxiliary nor primary task, nor their gradients provided a signal that could guide active learning. c) The mean of the Fisher Information of weights with respect to the loss function for Task 1 across 8 runs. The bars show one standard deviation. Fisher Information peak behaved differently in a deeper network with 3 convolutional layers, also not providing a signal to guide switching (mean of 12 runs, red vertical lines indicate the optimal switching window). d) Performance on task 2 for a deeper network with respect to amount of training on Task 1 and Task 2.

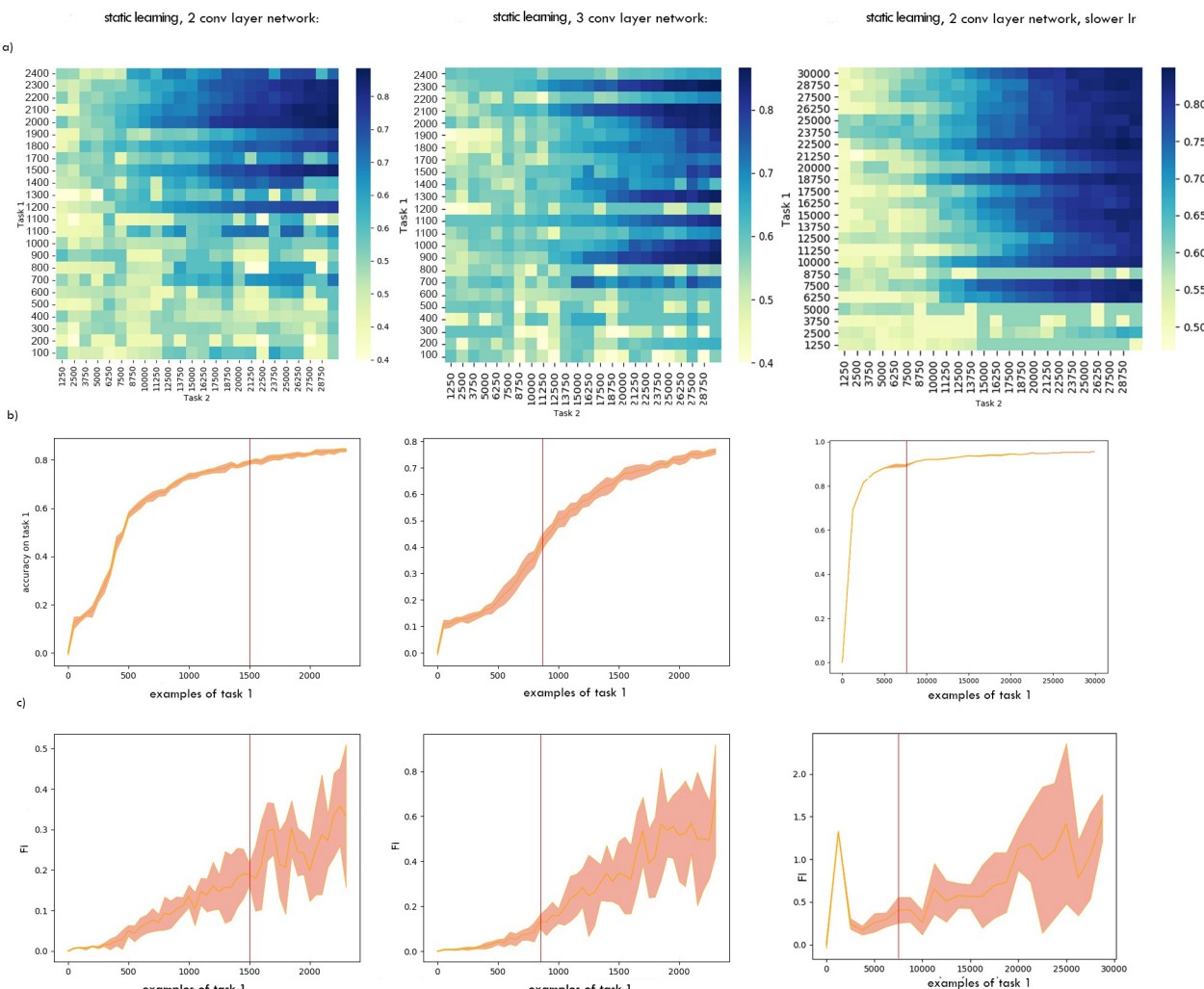

**Fig 6. Comparison of potential switching metrics during static learning.** The original 2-convolutional layer network, a deeper 3-convolutional layer network and the original 2-convolutional layer network with a slower learning rate of 0.0005 were compared. a) Performance on Task 2 for a with respect to amount of training on Task 1 and Task 2. b) Mean of 8 runs and standard deviation bands for performance on Task 1. c) Mean of 8 runs and standard deviation bands for Fisher Information.

**Table 1. The Fisher Information of weights for the original 2-conv network with respect to the Task 2 loss function during first 1250 examples of Task 2 training.**

| Amount of pre-training | FI in Task 2 | |
|---|---|---|
| | Mean | StDev |
| 2500 | 0.001124378395 | 0.002429203634 |
| 5000 | 0.0005223868013 | 0.002110925266 |
| 10000 | 0.001173843067 | 0.004293080181 |
| 15000 | 0.0008320442903 | 0.001386499149 |

We found no significant difference between various amounts of pre-training (p > 0.06 in all pairwise comparisons)

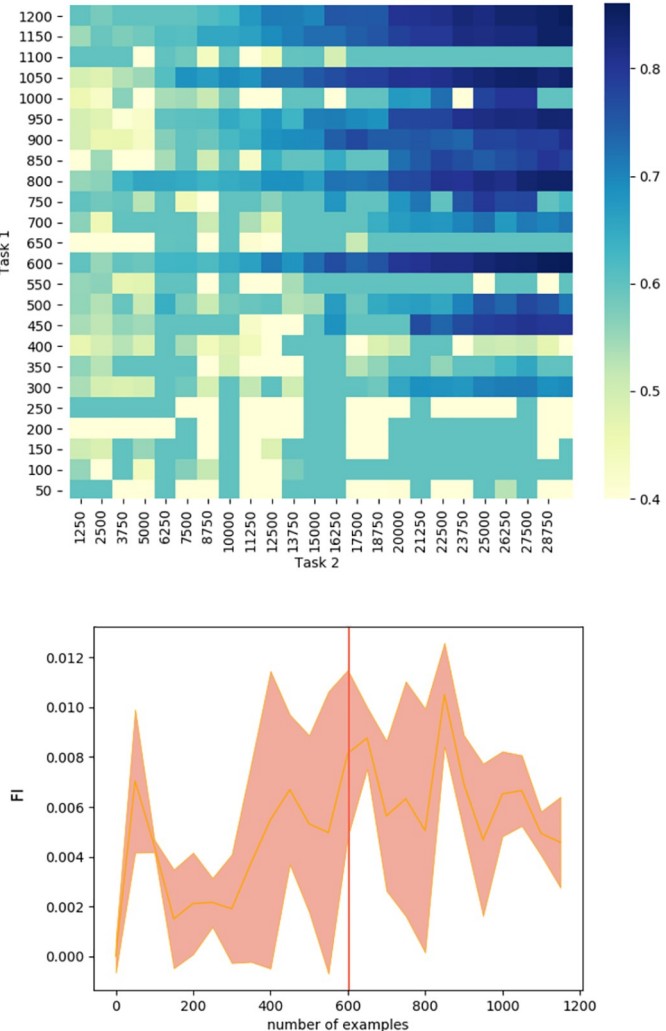

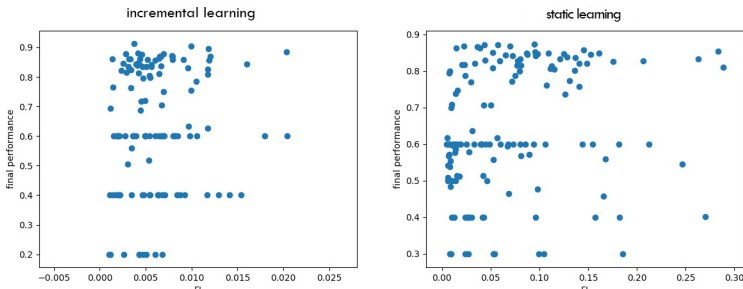

**Fig 7. Early stages of continual learning.** FI (mean of 6 runs and standard deviation) and final performance for smaller slices during the early stages of continual learning.

**Fig 8. Predicting final performance on Task 2 with FI.** Data from 8 runs, each point corresponds to the FI value for Task 1 at the moment of switching and the final performance on Task 2 for a single learning run. Correlation between final performance on Task 2 and Fisher Information with various degrees of training on Task 1: for continual learning: r = 0.115, p = 0.21, for static learning: r = 0.22, p = 0.007. In this case FI can only predict 1.3% and 4.8% of the Task 2 final performance.

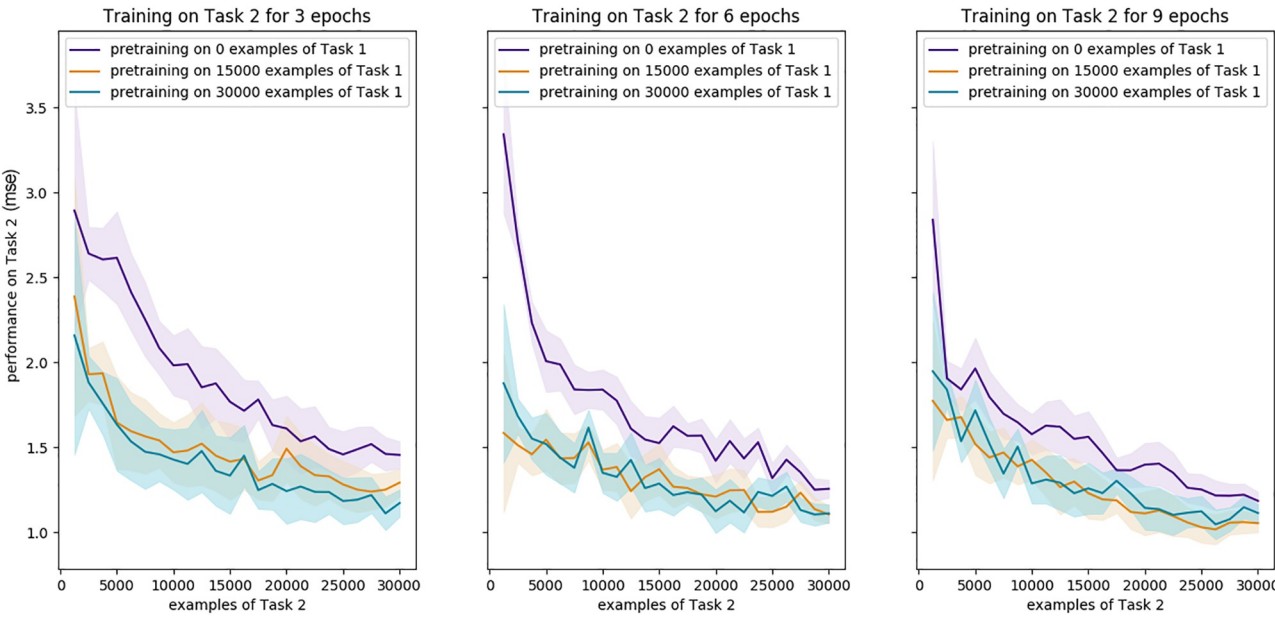

**Fig 9. Using FI to predict the gains from switching to another task on the next step.** FI for training on the next slice of Task 1 set and the first slice of Task 2 set after a given amount of pre-training on Task 1. After any amount of pre-training, information gain for continuing learning on Task 1 was significantly greater than for switching to the first slice of Task 2 set.)

### 3.3 Using digit addition for Task 2

In the odd-even task, the error signal was orthogonal to any single-image class, providing equal reward for learning and 'unlearning' it. Every digit was distributed equally in the odd and even digit-sum classes, and so learning weight changes after each batch of examples were cancelling each other.

An addition Task 2b, in which the network must report the sum of two digits in an input image, is in many ways similar to the odd/even task: two digits are presented; a mapping from an image to single representation would be valuable; and the output is derived from a combination of the two digits. However, in contrast to the odd/even task, it was possible to design it in a way so that the classes in Task 2b were not orthogonal to single digits. Our prediction, therefore, was that it would not require pre-training.

We found it could be learned without pre-training and differences in performance for varying amount of pre-training did not exceed the standard deviation between runs (Fig 10). This supports the interpretation that it is the relationship between the training labels and the desired intermediate representation (here, categorisation of single digits) that determines the need for pre-training.

### 3.4 Examining task representations

As performance for the addition task shows, these are likely to be dependent on the nature of the task. To understand how the representations developed in the network, we used t-SNE to see the manifold distances between a selection of digits. Fig 11 shows that examples consisting of two same digits proved to the most problematic to represent correctly. This suggests that active learning could be harder to perform on tasks with low inter-class distance.

The two tasks themselves were also orthogonal. Task 1, while providing the network with useful representations, did not give it any information about odd and even sums, and Task 2a

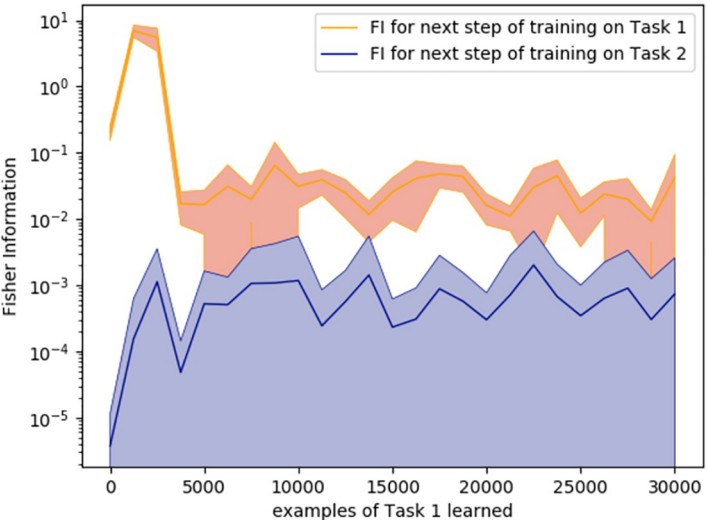

**Fig 10. Comparison of learning trajectories for the addition task with respect to various amount of pre-training.**
Mean of 4 runs and standard deviation bands for performance on Task 2 (mean square error is used instead of
accuracy for this task).

did not provide a useful signal for the network to be able to "catch up" with learning separate
digits if the representations were not already formed.

## 4 Discussion

This study highlights a limitation of automated curriculum learning in a simple task context. It
may not be a viable strategy for tasks in which the benefits of the chosen curriculum is not
apparent until much later. Such tasks would require an external curriculum to be learned
correctly.

Due to the limitations of architecture imposed by the original study goal, modelling true
non-stationarity and catastrophic forgetting was not attempted (S2 Fig in the Supporting
information can be referred to for more detail). However, even the simple case of balanced
class representation provided enough challenge to the current network when learning in a nat-
uralistic continual setting.

We tested four diverse types of measures, based on those that have proven successful in pre-
vious work. We found no evidence that any of the measures could indicate when sufficient
pre-training had been obtained, suggesting that active learning may not be possible for this
task. However, it should be noted that our search of measures was not exhaustive and it
remains possible that there is some statistic that would allow active learning in this case. A fur-
ther consideration is that even if there is some measure that would have worked in this task, it
may not be possible to know in advance which measure, and at what threshold, learning
should transition. Active learning would not be useful, if identifying its hyperparameters is
more difficult than determining the optimal curriculum through an exhaustive search.

We speculate that active learning will be particularly difficult when the loss function of the
pre-training and final task are orthogonal, so requiring quite different directions in gradient
descent. In such tasks learning one level, while providing useful representations for future
learning, will not teach the network anything about other levels. This should be especially
noticeable in case of specific kinds of multitask curricula. Consistent with this, we found that a

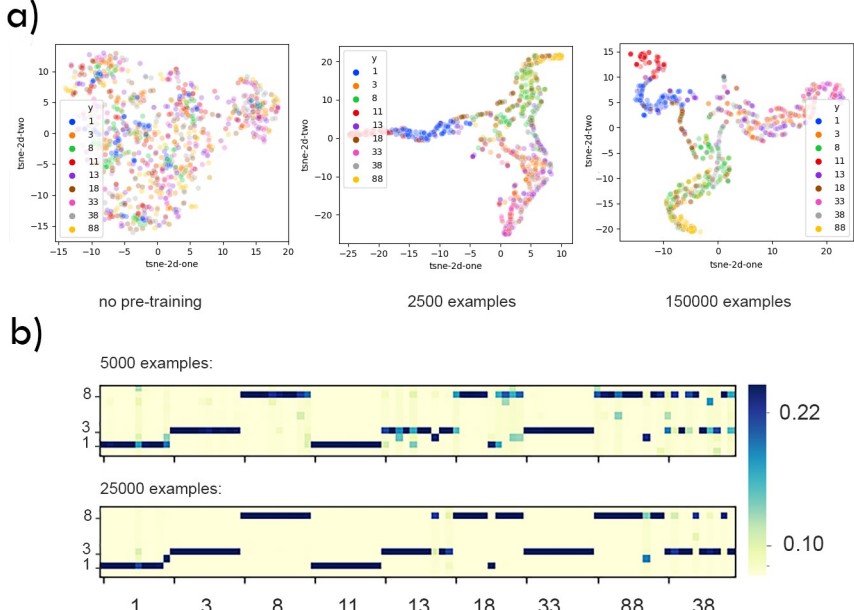

**Fig 11. Evolution of task representations with training.** a) t-SNE analysis of activations in the first dense layer of the CNN during the continual learning process. Representations of single digits are formed in the first few slices of the training set and change little with overtraining. The model was also able to discriminate repeated digits from single digits (e.g. "11" from "1"). However, the representation of images comprising two different numbers appears to be converging on the centre of the space: potentially, the "average" response to the two individual digits. b) First dense layer activation for training on 5000 and 25000 examples of Task 1. The x-axis shows the image classes presented to the network. Ten examples of each image class were shown. The y-axis labels the 10 neurons at the first dense layer, ordered by digit preference. Representations were stable over time. Single digits were reliably recognised, irrespective of the degree of training. However, when encountering stimuli consisting of two different digits it seems to only represent one of them, and this was exacerbated by overtraining.

addition task sharing many similar demands (Task 2b) did not benefit from pre-training as much as the odd/even digit sum task (Task 2a).

Ruder's overview [38] of multi-task learning approaches provides a summary of task relations. Simplifying his classification, we suggest that mutually assisting tasks (related or even adversarial tasks) may benefit from active learning. When tasks share features the model may have enough information to know when to switch. However, if one tasks provides a foundation for learning the other, but not the other way around, switching may prove problematic.

These results also pose a question for developmental robotics. It is possible that the challenges that active/curiosity-driven approaches encounter are not supposed to be solved by curiosity alone. For example, in educational practice it is accepted that there are benefit to an overarching framework to the curriculum that is fixed, but that within this active learning can be valuable [24]. This is supported by the Vygotskian concept of scaffolding, in which teachers support learners to achieve the next level by challenging them to the optimal degree, keeping them in their "zone of proximal development" [33]. Building upon these lessons from human learning, it seems likely that ANNs, as well as more complex robotic models, may too benefit from a combination of fixed curriculum and active learning.

## 4.1 Broader impact

Our work aims to use the study of human development and education to inspire new approaches to machine learning. It is a proof-of-principle work capturing the limits of

strategies for improving the performance of existing machine learning algorithms. It highlights potential directions for study in image recognition, natural language processing systems and reinforcement learning.

The second goal of our research is to use machine learning to provide a computational model of human learning. In particular, our group's goal is to understand learning in human infancy. As it is so difficult for us to imagine what it might be like to be an infant, computational models are valuable instantiations of the learning process. Predictions from these models can then be tested through the methods of developmental psychology and neuroscience.

## Supporting information

**S1 Fig. Effect of the number of epochs.** Doubling the amount of epochs made the task learnable, even if performance wasn't as good as with the addition of pre-training on Task 1. With the original amount of epochs performance on Task 2 without pre-training averages to random with respect to some noise. In some learning cases there was a pattern of cycling between 40% and 60% accuracy (we attribute this effect to classes where the same digit repeats twice, the result then depends whether the network associated the visual features of this digit with the odd or even class). Again, in multiple runs this averages to 50% accuracy.
(PNG)

**S2 Fig. Catastrophic forgetting scenario.** Class-incremental learning during pre-training led to a catastrophic forgetting scenario where the network never learned single digits representation enough to facilitate learning of Task 2. Due to the limitations of architecture imposed by the original study goal, ours isn't a suitable model to deal with true non-stationarity and catastrophic forgetting. Investigating the interplay between curriculum and active learning will be the direction of our future work.
(PNG)

## Author Contributions

**Conceptualization:** Anna Kravchenko, Rhodri Cusack.

**Data curation:** Anna Kravchenko.

**Formal analysis:** Anna Kravchenko.

**Funding acquisition:** Rhodri Cusack.

**Investigation:** Anna Kravchenko.

**Methodology:** Rhodri Cusack.

**Project administration:** Rhodri Cusack.

**Resources:** Rhodri Cusack.

**Software:** Anna Kravchenko.

**Supervision:** Rhodri Cusack.

**Validation:** Anna Kravchenko.

**Visualization:** Anna Kravchenko.

**Writing – original draft:** Anna Kravchenko.

**Writing – review & editing:** Anna Kravchenko, Rhodri Cusack.

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
