## [Decision Letter · Decision Letter 0]

13 May 2022

PONE-D-22-10095The limitations of automatically generated curricula for continual learningPLOS ONE

Dear Dr. Kravchenko,

Thank you for submitting your manuscript to PLOS ONE. After careful consideration, we feel that it has merit but does not fully meet PLOS ONE’s publication criteria as it currently stands. Therefore, we invite you to submit a revised version of the manuscript that addresses the points raised during the review process. Even though the three reviews differ in terms of recommendation (minor revise, major revise, reject), they all identify some merit and interesting contributions in the paper. However, they also identify important methodological weaknesses that will need to be addressed before the paper can be published. Please note that these weaknesses cannot be addressed with just changes in the writing/presentation -- instead, changes are required in the actual methods, selection of learning tasks, and experimental results. If you do not think that you can perform such substantial changes in the paper, it may be necessary to withdraw the paper from PLOS ONE. 

We look forward to receiving your revised manuscript.

Kind regards,

Constantine Dovrolis

Academic Editor

PLOS ONE

Journal Requirements:

[We thank our funders, the European Research Council Advanced Grant, FOUNDCOG 

(787981) and Science Foundation Ireland (17/RC-PhD/3482). We also thank Professor Geraldine Boylan at the INFANT Centre, University College Cork.]

 [This study was supported by the European Research Council Advanced Grant (https://erc.europa.eu/): 787981 (RC) and Science Foundation Ireland (https://www.sfi.ie/): 17/RC-PhD/3482 (RC). The funders had no role in study design, data collection and analysis, decision to publish, or preparation of the manuscript.]

Reviewers' comments:

Reviewer's Responses to Questions

**Comments to the Author**

1. Is the manuscript technically sound, and do the data support the conclusions?

Reviewer #1: Partly

Reviewer #2: No

Reviewer #3: Partly

2. Has the statistical analysis been performed appropriately and rigorously? 

Reviewer #1: Yes

Reviewer #2: N/A

Reviewer #3: Yes

3. Have the authors made all data underlying the findings in their manuscript fully available?

Reviewer #1: Yes

Reviewer #2: Yes

Reviewer #3: Yes

4. Is the manuscript presented in an intelligible fashion and written in standard English?

Reviewer #1: Yes

Reviewer #2: No

Reviewer #3: Yes

5. Review Comments to the Author

Reviewer #1: Summary

This paper tests four types of task-switching metrics to choose the optimal curriculum for continual learning. These metrics are namely, Performance, Competence, Prediction gain, and Information gain. The authors experiment with these metrics in a two-task scenario. The first task consists of standard MNIST digit classification. Then, the network presented a second task which requires summing two digits and deciding whether the sum is odd or even. The authors conclude that none of these metrics provided a signal that could guide active learning when training a network in a continual manner.

I think the overall paper is well-written and easy to follow. Furthermore, work is well-motivated based on the literature on human learning. However, I have some concerns regarding the choice of tasks. Also, I have concerns about the choice of network architecture and some of the experimental results. Please see below.

Comments

I suspect that Task2 does not necessarily require higher-level abstract information compared to Task1. Suppose our network can map 1,3,5,7,9 to binary value "1" and 0,2,4,6,8 to "0". Then Exclusive Or (XOR) operation on these two binary values will give us the desired output for Task2. Therefore, Task2 boils down to binary classification and learning the XOR function. So I can't entirely agree with "Task 2 required the network to recognize two digits simultaneously and analyse higher-level abstract information."

Experiments suggest that the network cannot learn Task2 alone. The final accuracy is 50% for binary classification, which is random. Authors suggest that this shows that this task structure was effective in requiring a curriculum. However, I firmly believe the network should be able to at least perform better than random without learning to classify digits. What is the authors' opinion regarding this issue? Based on my above comment, I strongly think the network does not necessarily need to know how to classify ten digits to perform Task2.

Based on figure-1, convolutional layers have only one filter. What is the rationale behind this choice since it is uncommon to have only one filter in convolutional layers? Figure 4 shows that the network reaches the accuracy of 80% on Task1 (MNIST classification). This is too low for the task. Simple two-layer MLP can reach accuracy values of around 97.5%. So I think the network may not have enough capacity to learn tasks due to one filter per layer.

Minor Comments

(1) A figure can support the explanation of the training procedure (section 0.3).

(2) In line 148, "n" is used for the first time, but the meaning of "n" is not explained.

(3) Numbering of sections is a little confusing. Specifically, all subsections have numbers like 0.X, which leaves an impression that all subsections belong to section 0.

(4) Some points in the Training procedure subsection are confusing to me. I think the main reason is the use of the word "batch". Specifically, I first thought that word "batch" referred to mini-batched used in stochastic gradient descent. However, later I interpreted word batch as mini-datasets that you used. So, I think it would help the reader if the meaning of batch is explicitly mentioned.

(5) I think there is a typo in line 172. zaremba should be \\cite{zaremba}

(6) Line 266 "Fig." is repeated twice.

Reviewer #2: Summary:

This study presents an interesting set of experiments that focus on detecting when a sufficient amount of learning has been done on one task to benefit another future task. The authors look at simple scenario of two tasks designed using the MNIST dataset. The primary experiments explore the impact of the training on the first task (task 1) before the second more complex task (task 2).

Despite this, I felt that some of the explanation of the experimental setup could have been more clear. The exact training scenario and how the data is presented to the models in a continual versus static way is not clear. The choice of hyper-parameters also do not seem ideal for this dataset and task and the authors provide no justification or experiments where these values are varied. This makes it difficult to comment further on the experimental results and discussion. My overall recommendation would be the following:

- make the training and evaluation protocol much more clear so that the results can be properly interpreted.

- ensure proper use of terminology, the use of the terms “active learning” and “continual learning” appear to be used quite loosely where I think (at least as it is currently described) the work is more similar to transfer learning.

- ensure that the network architectures, training times, and hyper-parameters are appropriate for the tasks to avoid drawing incorrect conclusions.

Individual Points:

The authors claim in section 0.1 that the goal was to design a sufficiently simple task such that they could exhaustively explore final performance across hundreds of different training scenarios, however the experiments seem fairly limited in terms of different scenarios outside of varying the amount of iterations over the data in the two tasks.

In section 0.3, the authors discuss both continual and static training in the context of two tasks, but I was not able to find a description of the exact distinction between the two training scenarios. Are the “sequential batches of 1250 examples” distinct in terms of the classes present or are they just subsets of the entire dataset randomly chosen. Is the continual part a distinction about freezing the weights for the task 1 part of the network? Understanding exactly how the networks are trained is crucial to evaluating the results.

In section 0.6 the authors claim that Task 2 was unlearnable without some initial training on task 1 and their result seem to indicate this. However, I am curious if this may be due to the training protocol or network architecture. Did the authors varying the length of training on either tasks (number of epochs) or vary the hyper-parameters? The addition of one conv layer in the task 1 portion of the network as mentioned in section 0.2 may not be enough, but I believe that task 2 should certainly be learnable with the right scenario without any pre-training.

Grammar:

Section 0.2, line 130 - "To investigate the generalisation of switching metric performance across networks, we also tested ---them--- a network variant that was similar, except that it had three rather than two convolutional layers." The word them should be removed.

Reviewer #3: Summary : The work aims at understanding the effect of curriculum learning with fundamental tasks are followed by tasks of higher complexity, in the context of continual learning or learning down stream tasks with higher level of complexity. The authors create two tasks, wherein the first task is a simpler task and can be used to solve the more complex task 2. The authors also experiment with 4 different metrics for deciding the optimal stage to switch between tasks ,that is how much pre-training on task-1 can give you the best performance on task-2. The authors conclude that an optimal amount of pre-training on task 1 improves the performance on task two, but over-training on task-1 leads to degradation in accuracy for task-2. Further, none of the metrics experimented with to decide optimal amount of pre-training turned out to provide valuable insight.

Comments :

1. I find this work really thought provoking, however the authors miss a lot more additional motivating factors. For example, curriculum learning or task evolution has been shown to promote modularity where the ANNs learn to solve tasks with increasing complexity.

2. I find the setting of the experiment very interesting but highly unclear. I would suggest to the authors to make the setting more clear.

For example, in the continual learning setting the authors write that the ANN observes a batch of size 1250 for 3 epochs and then does not observe those samples in further training. However, does that batch contain samples from all classes ?

If so then the claims about catastrophic forgetting made later in the results section may not be true. Clarity in the setting of experiment is highly lacking.

3. An experiment to show catastrophic forgetting by providing confusion matrices on both tasks and analyzing them would be beneficial. As the setting isn't clear this experiment may or ay not be useful.

4. I would also advise adding more tasks to the pipeline, for example the first task could be classification, then addition and then a task predicting addition modulo 2. This would help in understanding what is happening wen we try to scale curriculum learning. Further, I would also suggest very simple set of tasks, which can be easily proven to be hierarchical to foster more understanding than the t-SNE plots which vary highly with variables used to create them.

5. I am not convinced that the task and the architecture used are entirely correct.Is it so that when converting a single digit image to 84x84 the image is padded rather than resized ? If not, then is it so that augmentations are performed while training to give scale invariance to the ANN ? If both of those are not done then the 3x3 filters re not observing the same scale of inputs when moving from task 1 to task 2. This may hinder in the ability of ANNs to learn task 2 significantly.

6. Minor comment : The figures need to be captioned, labelled and of higher resolution.

6. PLOS authors have the option to publish the peer review history of their article (what does this mean?). If published, this will include your full peer review and any attached files.

Reviewer #1: No

Reviewer #2: No

Reviewer #3: No

---

## [Author Response · Author response to Decision Letter 0]

27 Sep 2022

Dear Professor Dovrolis,

Thank you and the reviewers for considering our submission so carefully. We are pleased to submit a revised draft, which has been substantially improved as a result of the comments.

As per your request, funding information has been removed from the manuscript.

Following your other comments, here is the correct information to add to the Financial Disclosure, Funding Statement and Competing Interests sections:

"This study was supported by the European Research Council Advanced Grant (https://erc.europa.eu/): 787981 (RC) and Science Foundation Ireland (https://www.sfi.ie/): 17/RC-PhD/3482 (RC)."

Best regards,

Anna Kravchenko

---

## [Decision Letter · Decision Letter 1]

19 Oct 2022

PONE-D-22-10095R1The limitations of automatically generated curricula for continual learningPLOS ONE

Dear Dr. Kravchenko,

Thank you for submitting your manuscript to PLOS ONE. After careful consideration, we feel that it has merit but does not fully meet PLOS ONE’s publication criteria as it currently stands. Therefore, we invite you to submit a revised version of the manuscript that addresses the points raised during the review process.

The reviewers (one of them in particular) have identified some minor points that need further improvement. Please address those thoroughly and the paper will be ready for publication.

We look forward to receiving your revised manuscript.

Kind regards,

Constantine Dovrolis

Academic Editor

PLOS ONE

Journal Requirements:

Reviewers' comments:

Reviewer's Responses to Questions

**Comments to the Author**

1. If the authors have adequately addressed your comments raised in a previous round of review and you feel that this manuscript is now acceptable for publication, you may indicate that here to bypass the “Comments to the Author” section, enter your conflict of interest statement in the “Confidential to Editor” section, and submit your "Accept" recommendation.

Reviewer #1: (No Response)

Reviewer #2: All comments have been addressed

Reviewer #3: (No Response)

2. Is the manuscript technically sound, and do the data support the conclusions?

Reviewer #1: Yes

Reviewer #2: Yes

Reviewer #3: Yes

3. Has the statistical analysis been performed appropriately and rigorously? 

Reviewer #1: Yes

Reviewer #2: Yes

Reviewer #3: Yes

4. Have the authors made all data underlying the findings in their manuscript fully available?

Reviewer #1: Yes

Reviewer #2: Yes

Reviewer #3: Yes

5. Is the manuscript presented in an intelligible fashion and written in standard English?

Reviewer #1: Yes

Reviewer #2: Yes

Reviewer #3: Yes

6. Review Comments to the Author

Reviewer #1: Summary

This paper tests four types of task-switching metrics to choose the optimal curriculum for continual learning. These metrics are namely, Performance, Competence, Prediction gain, and Information gain. The authors experiment with these metrics in a two-task scenario. The first task (pre-training) consists of standard MNIST digit classification. Then, the network is presented with a second task which requires summing two digits and deciding whether the sum is odd or even. The authors conclude that none of these four metrics provided a signal that could guide the learner to decide on optimal switching points.

I think the overall paper is well-written and easy to follow. Furthermore, work is thought-provoking and well-motivated. However, I have some concerns about claims, and a few details of the experiment setting are unclear to me.

Comments

General

In my opinion, the given evaluation scenario is closer to transfer learning than continual learning. Continual learning comes with several desiderata, such as avoiding forgetting, learning long sequences of tasks, and forward knowledge transfer. This work only focuses on the "forward knowledge transfer" aspect of curricula for continual learning.

It does not investigate forgetting, which is one of the most important problems in continual learning. For example, what if the metric "X" does not help predict the optimized switching moment in terms of accuracy on Task 2 but provides a switching moment that leads to less forgetting?

I see that this is briefly mentioned in Supporting information and Discussion. I would suggest clarifying this distinction in the introduction by highlighting that this work only focuses on one aspect of continual learning.

2.1 Data and tasks

My primary concern was that Task 2 does not necessarily require higher-level abstract information compared to Task1. The Authors partially address this concern with the "Orthogonality hypothesis" and a few empirical results. Also, I understand the motivation of keeping performance below the maximum possible accuracy by restraining computation resources (e.g., the number of epochs).

However, I think the claim "potentially even impossible to learn" (line 134) is inaccurate and needs to be revised. Suboptimal accuracy or requirement for more resources differs from failing to learn. Furthermore, in "S1 Fig.1" the authors show that Task2 is learnable (but with worse accuracy) if the number of epochs is changed. I strongly believe that Task 2 can even be learned with fever epochs with higher accuracy with a more optimal training routine (e.g., learning rate schedule, weight decay, proper batch size).

2.2 Model

(1) In the first sentence, the number of filters of convolutional layers is not given. I think it should be also presented here (I see it is given in Figure 1) to understand the model's learning capacity better.

(2) I think the CNN visualization in Figure 1 is confusing. For example, the input size and the number of filters are given as dimensions of the same shape. In deep learning literature, it is more common to have LeNet or AlexNet style visualizations -- please see https://alexlenail.me/NN-SVG/LeNet.html and https://alexlenail.me/NN-SVG/AlexNet.html.

(3) The authors propose to train output layer-1 for task-1 and a second output layer that follows output layer-1 for task-2. I think a justification for this choice should be provided because it is not common to have such output structures in continual or transfer learning. Specifically, either independent task output heads are used for different tasks (e.g., continual learning multi-head scenario), or the previous task output layer is replaced with a new one (e.g., transfer learning with different targets).

2.3 Training procedure

Some hyperparameters (e.g., optimizer, weight decay, learning rate) and how they are determined are unclear. I suggest that the authors mention such details in this subsection.

Supporting information

(1) S1 Fig.1 Effect of the number of epochs. Legend is missing in the plot. Does the blue line double the number of epochs?

(2) S2 Fig.1 Catastrophic forgetting scenario. The sentence "Class-incremental learning during pre-training learned to a catastrophic forgetting scenario that couldn’t have been solved simply with switching at the correct time." is not clear to me. I suggest that the authors rephrase this part.

(3) Furthermore, the term "Class-incremental learning" is used for a specific continual learning scenario where the learner encounters novel classes in a sequential task and requires predicting all the previous classes regardless of task knowledge. However, if my understanding is correct, all ten digits are presented to the learner during pretraining. So, it is not a class-incremental learning scenario.

Reviewer #2: My major concerns from the original submission have been addressed and the work now has a clear description of the experimental protocol. The design of the experiment still seems a bit weak to me, but it does still support the conclusions found by the authors. Overall, the work is ready to be accepted.

Reviewer #3: I would like to thank the authors for addressing all the previous concerns. The writing has been improved drastically and many more details have been added ( which were lacking ) making the manuscript more thorough.

The only concern that remains is that of lacking evidence regarding the conclusion that amount of training on Task-1 has a drastic effect on Task-2. Although I agree with the main experiment conducted in the work, I would also like to see more evidence for the same through another experiment with different tasks but with increasing complexity. Example : MNIST digit classification can be Task-1, and the Task-2 can be class specific rotation, where depending on the class the input image is rotated by some fixed angle. Class-1 : angle 30, Class-2 : angle 45 etc.

7. PLOS authors have the option to publish the peer review history of their article (what does this mean?). If published, this will include your full peer review and any attached files.

Reviewer #1: No

Reviewer #2: **Yes: **Cameron Taylor

Reviewer #3: No

---

## [Author Response · Author response to Decision Letter 1]

9 Aug 2023

Dear Dr. Dovrolis,

This manuscript has been previously sent back for minor revisions. We thank you and the reviewers for your critical assessment of our work and we hope that the current version meets the standards of PLOS ONE.

Kind regards,

Anna

---

## [Editor Report · Decision Letter 2]

15 Aug 2023

The limitations of automatically generated curricula for continual learning

PONE-D-22-10095R2

Dear Dr. Kravchenko,

We’re pleased to inform you that your manuscript has been judged scientifically suitable for publication and will be formally accepted for publication once it meets all outstanding technical requirements.

Kind regards,

Constantine Dovrolis

Academic Editor

PLOS ONE
---

## [Editor Report · Acceptance letter]

25 Aug 2023

PONE-D-22-10095R2 

The limitations of automatically generated curricula for continual learning 

Dear Dr. Kravchenko:

I'm pleased to inform you that your manuscript has been deemed suitable for publication in PLOS ONE. Congratulations! Your manuscript is now with our production department. 

Kind regards, 

on behalf of

Dr. Constantine Dovrolis 

Academic Editor

PLOS ONE